# The Viral Capsid: A Master Key to Access the Host Nucleus

**DOI:** 10.3390/v13061178

**Published:** 2021-06-20

**Authors:** Guillermo Blanco-Rodriguez, Francesca Di Nunzio

**Affiliations:** 1Advanced Molecular Virology and Retroviral Dynamics Group, Department of Virology, Pasteur Institute, 75015 Paris, France; Guillermo.blanco-rodriguez@curie.fr; 2Immunity and Cancer Department, Curie Institute, PSL Research University, INSERM U932, 75005 Paris, France

**Keywords:** viral nuclear entry, viral replication, host–viral interactions

## Abstract

Viruses are pathogens that have evolved to hijack the cellular machinery to replicate themselves and spread to new cells. During the course of evolution, viruses developed different strategies to overcome the cellular defenses and create new progeny. Among them, some RNA and many DNA viruses require access to the nucleus to replicate their genome. In non-dividing cells, viruses can only access the nucleus through the nuclear pore complex (NPC). Therefore, viruses have developed strategies to usurp the nuclear transport machinery and gain access to the nucleus. The majority of these viruses use the capsid to manipulate the nuclear import machinery. However, the particular tactics employed by each virus to reach the host chromatin compartment are very different. Nevertheless, they all require some degree of capsid remodeling. Recent notions on the interplay between the viral capsid and cellular factors shine new light on the quest for the nuclear entry step and for the fate of these viruses. In this review, we describe the main components and function of nuclear transport machinery. Next, we discuss selected examples of RNA and DNA viruses (HBV, HSV, adenovirus, and HIV) that remodel their capsid as part of their strategies to access the nucleus and to replicate.

## 1. Nuclear Transport Machinery

The cytoplasmic and the nuclear bi-directional exchange of molecules and proteins is ensured by the Nuclear Pore Complexes (NPCs). These structures are large multiprotein complexes composed by approximately thirty different nucleoporins in mammalian cells [1] organized in an eightfold rotational symmetry [2,3]. The development of electron microscopy techniques has provided a wide range of information about the structure and organization of the NPC. Particularly, in recent years, the advent of cryo-EM (electron microscopy) tomography [4,5] allowed high-resolution imaging at a nearly physiological state to obtain a clear view of the NPC structure. Thanks to these techniques, we obtained for the first time a detailed model of the human NPC with high resolution of ≈66 Å [6,7,8]. Even more details have been obtained by cryo-EM and single particle analysis, reaching a new primacy in resolution of 15 Å for the NPC of *Xenopus laevis* [9]. The NPC is composed by three different rings: the cytoplasmic ring, the intermembrane nuclear envelope ring, and the inner nuclear ring. The cytoplasmic and the nuclear rings have protrusions composed by different nucleoporins that form the cytoplasmic filaments and the nuclear basket [10]. The NPC architecture and density may vary between species [11]. Indeed, the nucleus of the mature Xenopus oocyte contains roughly 60 NPCs/µm^2^ and ≈5 × 10^7^ NPCs/nucleus in total [12]; this nucleus is almost 4 log more dense in NPCs in comparison to the nucleus of a human cell [13]. The human NPC has an outer diameter of around 1200 Å, an inner diameter ≈ 425 Å, and a height of 800 Å with a total mass of 110 MDa [14,15].

Two mechanisms determine the passage of molecules through the NPC: facilitated and passive diffusion. Macromolecules lacking nuclear transport signals (NLS), which create a specific amino acid sequence, are transported in the nucleus by passive diffusion. The structure and organization of the NPC allows the passive diffusion of small molecules and ions smaller than 9 nm. On the other hand, molecules larger than 39 nm require an active process to be imported or exported through the NPC [16]. However, recent studies highlighted that the NPC channel is flexible and allows the passive diffusion of much larger molecules than previously assumed (up to 230 kDa in size) at rates that decrease with increasing mass [17]. The central channel of the NPC [17,18,19] is decorated by several FG-Nups that in coordination with other Nups located within the inner ring complex, Nup170 and Nup188 [20], altogether contribute to maintain the permeability properties of the passive diffusion channel. Contrary to the passive diffusion, the facilitated (active) transport is mediated by nuclear transport factors (NTFs) [17]. Most frequently, the import of large cargos is achieved with the help of a family protein named β-Karyopherins [21,22,23]. Many proteins are imported via two subclasses of β-Karyopherins denominated importin-α and importin-β, and some other proteins are imported by alternative β-Karyopherins called transportins [24]. Cargo proteins with an NLS are recognized by the carrier protein α-importin, which is engaged by β-importin, forming an heterodimer [21,22,23]. Then, the cargo protein associated with the α/β importin heterodimer penetrates the NPC and engages in low-affinity interactions with nucleoporins containing FG repeats. This central region of the NPC contains approximately ten nucleoporins with several phenyl-alanine–glycine domains that create an unstructured mesh inside the NPC [25]. This mesh of disorganized FG repeats is responsible for the permeability of the NPC. Several models have tried to link the physical properties of FG repeats to the selectivity of the central channel of the NPC. For example, the “Forest” model proposed by Yamada et al. [26] suggests that the FG selectivity is achieved by the combination of FG repeats with different structure and chemistry properties. The “Forest” model is characterized by the collapsed-coil domains that form a transport zone 1 in the middle of the NPC, which is surrounded by a zone 2 formed by the extended-coil domains. Small molecules, transport receptors, and complexes can translocate through zone 1 or 2 based on their size, surface charge, and hydrophobicity. Alternatively, the virtual gate model proposed by Rout et al. [27] considers that the selectivity of the NPC is achieved by entropy. According this model, the random movement imposed by the size of large molecules limits their passage through a confined space such as the NPC channel [28]. Consequently, the impediment for the cargo translocation through the NPC is imposed by a physical and entropic barrier rather than a selection imposed by the FG structure. Similarly, Frey et al. observed that the gel-like properties generated by the FG repeats might be enough for imposing a diffusion barrier to the NPC [29]. Other works suggest that the selectivity of the NPC is controlled by the concentration of karyopherins [30]. According to these observations, karyopherins act as “molecular Velcro” made from intrinsically disordered domains that reduce the diffusion of large molecules through the NPC. Another recent model suggests that the FG repeats impose a constraint to the passive diffusion of molecules, and only small molecules are able to overcome this barrier [31]. Nevertheless, when large cargoes are transported by karyopherins, they can penetrate the NPC thanks to the interaction between karyopherins and the FG repeats of Nups that collapse. Then, this collapse is reversed by Ran guanosine triphosphate returning the FG to a polymer brush-like state [31].

In addition, several molecular dynamics simulations have been extensively performed over the past decade to shed light on the transport mechanism. Usually, simulations are conducted over a time scale of micro seconds to determine how FG Nups stretch out and form rapid bundles of multiple FG Nups. It has been proposed that the hydrophobic region of FG Nups is protected from being exposed to the aqueous environment by being coiled up into a globular structure [32,33,34].

A more recent study highlights that FG-nucleoporins from different Nups undergo liquid–liquid phase separation and form liquid droplets that mimic the permeability barrier properties of NPCs [25,35]. In addition, the directionality of the transport of cargoes across the NE is ensured by the small GTPase Ran. The chromatin-associated factor, RCC1, is the RanGTP exchange factor (RanGEF) that maintains a high RanGTP concentration in the nucleus. At the cytoplasmic side of the NPC, high RanGDP levels in the cytosol are ensured by the Ran GTPase activating protein (RanGAP) that binds to RanBP2/Nup358 [36,37].

So far, we have reached a good understanding of how proteins are imported to the nucleus, but much less is known about the strategies and tactics that viruses employ to cross the nuclear barrier. In the next sections, we expose some examples of how different viruses have evolved alternative ways to access the host nucleus. We will particularly focus on the role of the capsid to lead the viral genome into the host chromatin compartment, which is an essential step for the life cycle of several viruses.

## 2. HBV

The hepatitis B virus (HBV) is an enveloped DNA virus with a size of 42–47 nm that belongs to the hepadnaviridae family [38]. Inside the capsid is enclosed a partially double-stranded DNA of 3.2 Kb adopting a relaxed circular shape. This organization of DNA is denominated rcDNA and contains four overlapping reading frames. HBV penetrates in the hepatocytes through a low-affinity interaction between the viral envelope glycoproteins and the cellular surface heparin sulfate proteoglycans [39]. Then, the preS1 region of the large envelop protein (L) interacts with high affinity with the HBV main receptor, the hepatic bile acid transporter sodium taurocholate co-transporting polypeptide (NTPC) [21]; successively, the virus is internalized via caveolin-1 or clathrin endocytosis [40,41]. Then, the HBV capsid is directed to the nucleus through microtubule interactions [42]. The viral genome translocates into the nucleus of an infected cell through the capsid. The HBV capsid is composed by repeated units of the core protein, which is a small protein of 185 amino acids with a molecular mass of 21 kDa. The core protein has two NLS signals in its C-terminal domain [43,44,45], which are exposed only after phosphorylation [46]. Then, after phosphorylation, the capsid binds and crosses the NPC [46], remaining assembled [16]. The HBV capsid, in order to cross the NPC, first interacts with the importin-α and β and then with Nup 153. Nup153 engages the capsid and by an unknown mechanism triggers the uncoating of the viral capsid [47] (Figure 1A). The nuclear import of the HBV capsid is a tightly regulated process, since only the mature capsid is able to cross the NPC and release the viral genome. On the other hand, immature capsids are also able to penetrate the NPC [48]. However, they remain blocked at the nuclear basket of NPC, as shown by experiments in permeabilized cells [48]. Inside the nucleus, the rcDNA is processed by the cellular machinery, converting it to covalently closed circular DNA (cccDNA) [49]. The cccDNA is linked to histones H3 and H4, creating a viral minichromosome [50] that serves as a template to produce the pregenomic RNA (pgRNA) and subgenomic RNAs. The pgRNA is exported to the cytoplasm to serve not only as template for the translation of the viral core protein and polymerase but also for the generation of new virions. The pgRNA encapsidation is triggered by its binding with the polymerase. This binding renders the formation of immature nucleocapsids in which the reverse transcription of pgRNA into minus strand DNA takes place. The immature HBV capsids are covered by the viral envelope proteins in the ER and expulsed through the cellular secretory pathway [51].

The viral capsid is usually a key player in the viral infection, especially in viruses that need to access the nucleus to replicate. It is not surprising that the HBV capsid requires a change in its structure to meet the requirement of crossing the NPC, which is the main bottleneck for several viruses to complete their life cycle. Thus, the HBV remodeled capsid kidnaps cellular factors to access the nucleus. Observations showing that the capsid should be phosphorylated to change the conformation and interact with Nup153 highlight the importance of capsid remodeling and its interplay with nuclear pore factors to ensure the viral nuclear entry.

## 3. HSV-1

The herpes simplex virus 1 (HSV-1) is a virus that belongs to the herpesviridae family. Its prevalence in the human population is high [52] and frequently causes a long-life infection characterized by long latency period and occasional reactivations [53]. The HSV-1 is an enveloped virus with an icosahedral capsid that enclosed a genome of dsDNA of 152 Kb. Between the capsid and the envelope, there is a protein layer called tegument [54]. The tegument occupies two-thirds of the virion volume and is composed by approximately twenty-three viral proteins in addition to viral and cellular transcripts [55,56,57]. Interestingly, as revealed by Cryo-EM studies, the capsid is not centered in the virion but displaced to an extreme, near the viral envelope [58].

The HSV-1 enters into the cells assisted by the proteins located in the viral envelope. The protein gB is mainly responsible, together with other proteins (gD, gH/gL), for interacting with the cellular receptors and forming the core fusion machinery [59,60,61,62]. Once the virus fuses with the host membrane, the core is released in the cytoplasm associated to the tegument proteins. Next, the viral journey is accompanied by microtubules and dynein toward the NPC [63,64,65]. Eventually, the viral capsid and the remaining tegument proteins reach the NPC. Their binding triggers the delivery of the viral DNA to the nucleus helped by the high pressure that exists inside the viral core [66].

The HSV-1 capsid shields the viral DNA in a space of 125 nm of diameter in mature angular shells (triangulation number T = 16) [67,68]. The capsid is composed of 4000 protein subunits organized in hexons, pentons, and triplexes [67,69]. On the other hand, the tips of the exon are crowned by the VP26 protein. The role of the capsid in the delivery of the viral DNA in the host nucleus is essential. This structure intervenes in two steps during the early time post infection: for the transport of the virus through the microtubules and for the viral nuclear entry through the NPC [63,64,65]. It has been observed that Nup358 and Nup214 are mainly responsible for the docking of the HSV-1 capsid at the NPC [70] (Figure 1B). Indeed, the depletion of Nup358 and Nup214 determines a reduction of the HSV-1 capsids at the NPC [71,72]. Both nucleoporins, Nup358 and Nup214, are exclusively located in the cytoplasmic side of the NPC and interact with the viral proteins UL36 and UL25 [73,74,75]. Experiments employing temperature-sensitive mutants of UL36 and UL25 revealed that these two proteins not only are able to bind the nucleoporins, but they are also involved in the release of the viral DNA in the nucleus. It has also been observed that the docking of the HSV-1 capsid at the NPC occurs in a specific orientation in which a capsid vertex is oriented toward the pore [76]. This portion of the capsid contains ten copies of UL25 protein that seem important for an efficient delivery of the viral DNA in the nucleus [77]. However, the entire molecular mechanism triggering the release of the viral DNA is not well understood. It has been hypothesized that UL36 and UL25 are probably the main effectors of the process. Nevertheless, it is known that the high DNA pressure inside the viral capsid, estimated to be ≈20 atmospheres [66], is responsible of the viral DNA ejection (Figure 1B). Of note, HSV-1 empty ghost capsids docked at the NPC are observed only when matured virions have ejected the DNA into the nucleus as the EM pictures show [70]. Interestingly, similar empty nearly intact cores have been recently found in cells infected with HIV-1 by cryo-EM studies near the NPC but inside the nucleus [78]. These results suggest a common mechanism of viral DNA ejection among divergent viruses to import their genome inside the host chromatin compartment. However, the mechanism that induces the release of the viral genome due to an intra-capsid pressure can be similar among HSV-1 and HIV-1, even if this phenomenon at least for HIV-1 remains so far unclear.

The HSV-1 strategy to access the nucleus represents another example of different adaptations to overcome the cellular barriers. As we have highlighted, the capsid of HSV-1 is a main player in the infection process and is fundamental to gain access to the nucleus. Similar to other viruses, the interplay between the capsid and the components of the NPC provides adequate conditions for the preservation and nuclear translocation of viral genome. Interestingly, the EM techniques as well as host–viral interaction studies on HSV-1 have been instrumental for increasing our knowledge about the HSV-1 nuclear import [67,68]. Biochemical data have been complemented in the last years with new results obtained from EM techniques, which highlight the HSV-1 capsid structure and its remodeling. 

The plethora of new and classical technologies will contribute altogether to provide new pieces of information on the process of HSV-1 nuclear import.

## 4. Adenovirus

The adenoviruses are a family of viruses that can infect mammals, birds, and even fish. The adenoviruses that infect humans are divided in different categories named A, B, D, E, F, and G and 57 different serotypes [54]. Among them, the most studied are serotypes 2 and 5. Adenoviruses can produce diverse pathologies or even be asymptomatic. They most frequently replicate in the eyes, in the respiratory, and in the gastrointestinal tracts [54]. The adenoviruses are non-enveloped viruses carrying a genomic material composed by ≈35 Kb dsDNA that codes for approximately forty proteins [79]. The dsDNA of the virus is highly condensed by the viral proteins V, VII, and X that help to organize the virion structure [54]. The capsid is mainly composed by 720 subunits of the hexon protein organized in 240 trimers, but other proteins are also required to form the correct capsid configuration [80]. The structure of the capsid is an icosahedron composed by 20 facets, each composed by 12 hexon trimers and a penton at each vertex [80]. Additionally, five penton base (PB) monomers are located on each of the icosahedral vertices. Finally, the tip of the penton is crowned by the fiber protein assembled in a trimeric fashion [80]. This fiber protein is the main protein responsible for engaging the cellular receptors that grant the access to the cellular cytoplasm. Adenoviruses use several different receptors and low-affinity interactions to penetrate the host cells. Adenoviruses that cause respiratory, ocular, or gastrointestinal pathologies use the coxsackievirus AdV receptors (CAR). On the other hand, adenoviruses that infect renal and respiratory tissues use desmoglein-2 receptor (DSG2) [81]. Once the virus interacts with its receptor, it is internalized by endocytosis mediated by the clathrin [82]. During this process, the fiber proteins at the tip of the capsid penton dissociate, and this process continues further in the endosomal internalization and during the escape [81]. Adenoviruses escape from the endosomes using the amphipathic helix of protein VI to disrupt the membrane of the endosome, which is likely helped also by a drop of the pH [81,82]. Then, the partially assembled capsid of the virions engage the dynein and microtubules through the interaction with hexon or proteins VI [83,84]. These interactions promote the docking of the virus at the NPC. Then, the viral hexon protein binds the Nup214. This interaction is determinant for the completion of the capsid uncoating and for the release of the viral genome into the host nucleus (Figure 1C). The intact adenovirus capsid with a diameter size ≈ 86 nm [85] is too large to efficiently fit into the nuclear pore channel (≈39 nm) [16]. Furthermore, the uncoating process of adenoviruses begins as soon as the virus penetrates the cell [81,82]. As mentioned before, the adenovirus hexon protein is able to interact with Nup214 [86]. It has been shown that antibodies against the N terminus of Nup214 prevent the docking of adenovirus capsid at the NPC [86]. In fact, the region of Nup214 responsible for the binding to the adenovirus hexon protein has been mapped in a sequence of 137 amino acid in the N-terminal domain. Cassany et al. [87] showed that the presence of Nup214 is required for efficient nuclear entry and capsid disassembly. On the other hand, the importins α and β are not required by the virus to deliver its DNA to the nucleus. However, another component of NPC, the Nup358, is also required for the capsid disassembly [88]. The proposed mechanism involves the binding of the adenovirus capsid to Nup214 and to the light chain of kinesin-1. Kinesin-1 is also attached to the NPC via Nup358 but through its heavy chain [88]. It has been hypothesized that the binding of the capsid to the aforementioned Nups enhances the nuclear import of adenoviruses. Finally, the depletion of kinesin-1 and Nup358 impairs the viral infection [88], which is facilitated by the core protein VII of adenoviruses that remains associated to the viral DNA after the capsid disassembly [84]. It has also been found that transportin-1 facilitates the nuclear access of the adenoviral protein VII [89], but it is probably not the only factor involved in the nuclear import. In fact, several nuclear factors have been linked to the nuclear import of the viral DNA such as hsp70, Histone H1, importin β, importin 7, transportin 1, and Mib1 [86,89,90,91]. Taken together, these results indicate that the adenovirus capsid is an essential structure for the infection. The capsid covers multifaceted roles in different steps of the infection including the entry into the cells, the journey toward the NPC, and the nuclear import. Similarly to other viruses, the timing and regulation of the adenovirus uncoating is also crucial for a productive infection, since the presence of hexon proteins is required for NPC interaction. Furthermore, the partially assembled capsid of adenovirus is probably also important to avoid the detection of viral dsDNA by antiviral sensors such as cGAS. Overall, the capsid and its structure remodeled during the cytoplasmic journey are critical for the adenoviral infection.

## 5. HIV

The human immunodeficiency virus (HIV) is an RNA virus that belongs to the retroviridae family. This family of viruses has the particularity of reverse transcribing the RNA genome into DNA, which is the only viral form capable of integration into the host DNA [92,93]. The HIV genome is ≈9.7 kb in length and codes for nine different ORFs, which are translated in fifteen different proteins [94]. The viral genome is protected by the core (120 nm in height, 40 nm and 60 nm in length) that occupies the larger part of HIV virions, which is composed by repeated units of the CA protein (≈1500) [95,96,97]. In order to form the characteristic HIV conical core, the CA monomers are assembled mostly in hexamers (around 250) and some pentamers (approximately 12). The presence of pentamers is required to provide enough structural flexibility to complete the conical structure of the core [98]. To penetrate the cells, the HIV engages low-affinity interactions with the cell surface in the search for the CD4 receptor using the Envelope glycoprotein (Env), which is arranged on the surface of the virus as a heterotrimer. Each monomer is formed by the receptor-binding surface unit (gp120) and the fusogenic gp41 transmembrane unit. The gp120 engages the CD4 receptor, triggering a conformational change that exposes a new region of the protein to be able to interact with the co-receptor CCR5 or CXCR4. Next, the gp41 catalyzes the fusion of target and viral membranes. After viral-cell fusion, the viral core is released in the cytoplasm and engages the microtubules to travel toward the NPC [99,100]. In particular, HIV recruits BICD2 (dynein adapter) and Fez1 (kinesin adapter) to progress toward the NPC helped by the dynein–dynactin complexes [101]. During its journey toward the NPC, the retrotranscription of the viral RNA to DNA starts inside intact cores [102,103]. In order to reach the host chromatin for integration and replication, HIV-1 must overcome the NPC barrier. However, the HIV core is too large to cross the NPC completely assembled. According to several lines of evidence, the maximum size of cargo that can cross the NPC is 39 nm [16], whereas the size of the assembled HIV core is larger [95,96,97]. Therefore, the core must disassemble at least partially or remodel to access the nucleus. For many years, the viral core uncoating (complete loss of viral CA) mechanism was considered to occur immediately after viral entry into the cell. This was the predominantly accepted mechanism. In recent years, the concept of the core uncoating has evolved. However, how the virus dismantles the core and the signals that trigger this event are still open questions that are currently investigated by several laboratories around the world. Different cellular compartments hosting the viral uncoating have been proposed: the cytoplasm, the NPC, or the nucleus. Divergences between cells could explain a different privileged path of uncoating than another [71,98,104]. Recent studies suggested that intact or nearly assembled capsids can penetrate the NPC [102,105,106]. This hypothesis is supported by the evidence that the nuclear entry of the viral capsid is orchestrated by its binding with several Nups [107,108,109,110]. Among them, the main Nups are RanBP2/Nup358 (a cytoplasmic filament protein of NPC) [111] and Nup153 (a nuclear basket component of NPC) [108,109,110]. Another novel notion in the field consists in the nuclear uncoating [78,112] and formation of a mature pre-integration complex (PIC) into the nucleus [113,114]. Once the virus reaches the nuclear environment, it interacts with host factors that determine the viral fate, such as the cleavage and polyadenylation specificity factor subunit 6 (CPSF6) and the lens epithelium-derived growth factor (LEDGF) [115,116]. Overall, the HIV-1 capsid plays multifaceted roles during infection. However, because of its inherent fragility [117], previous biochemical studies could not fully reveal the key role of the capsid in viral infection [118]. One of the first roles of this polymorphic structure is to provide a shield against the antiviral cellular sensors [119,120,121] and restriction factors, such as Trim5α and MX2 [122,123]. The viral core acts also as a protective environment for the reverse transcription (RT), which is the process that converts the viral RNA into DNA. The RT is a process linked to the core stability. Indeed, when the reverse transcription is blocked by inhibitors or by particular mutations in the reverse transcriptase enzyme, the uncoating is delayed [104,124]. Interestingly, hyperstable core mutants are defective for the RT [125]. Only recently has it been possible to directly observe that the nuclear import can precede the completion of reverse transcription and uncoating. Both processes were previously considered exclusive cytoplasmic steps. In particular, the susceptibility of the capsid to PF74 following nuclear import reveals that the assembled or partial capsid mediate post intranuclear entry steps [126,127,128,129]. Nevertheless, at least a partially assembled core is necessary to engage the cellular partners required for the viral nuclear import; indeed, Nup153 and CPSF6 bind to CA hexamers and not to CA monomers [130,131]. Of note, the majority of the interactions of the cellular factors with the viral capsid occur through a hydrophobic pocket formed between CA hexamers. This implies that at least some portions of the capsid must remain assembled to permit this interaction. Nup153, the most dynamic Nup, located in the nuclear side of the NPC, assists the nuclear translocation of HIV through its interaction with the viral capsid at least partially assembled (Figure 2).

The relevance of Nup153 in HIV nuclear import has been demonstrated by Nup153 depletion experiments. The knockdown of Nup153 severely affects the HIV nuclear import [108,109,132]. Other factors such as TNPO3 (part of nuclear import machinery of the cells) and CPSF6 have been implicated in the nuclear import of the virus to the nucleus [133,134,135]. CPSF6 competitively binds the same hydrophobic pocket of the capsid of Nup153, displacing the latter and releasing the viral capsid in the nucleoplasm [136]. This displacing of viral capsid partners could be also induced by the remodeling of the viral capsid during the nuclear translocation [106]. CPSF6 is able to interact only with the hydrophobic pockets formed between hexamers [130,131]. Therefore, at least partially assembled core fragments must be present to engage CPSF6 [106]. Of note, recent discoveries shine new light on the post nuclear entry steps and on the role of CPSF6 in viral replication. CPSF6 contains intrinsically disordered mixed-charge domains that can trigger liquid–liquid phase separation (LLPS) [137]. We recently observed this phenomenon induced by HIV infection [113,114] (Figure 3).

LLPS gives rise to membraneless organelles (MLOs) that we found in the nucleus of infected cells triggered by viral infection; in particular, these MLOs are formed by CPSF6, viral RNA genome, viral integrase, viral CA, and nuclear speckle factors, while the host chromatin is excluded from them [113,114]. It is interesting to note that the discovery of HIV-MLOs as sites of nuclear RT and of PIC maturation [113,114] obliges the entire scientific community to revise archived notions on the early steps of HIV-1 infection. These discoveries open up countless new questions. We could ask whether this phenomenon is related only to the biology of HIV-1 or if it can also be extended to other retrovirus family. There is currently a highly active debate about the state of the HIV capsid during nuclear import. Recently, we observed cores apparently intact docked at the NPC [106], which was confirmed with a better imaging resolution by Zila et al. [78]. CA gold labeling showed a different distribution in the viral complex between the cytoplasm and the nucleus, suggesting different morphologies [106]. Notably, when HIV-1 reverse transcription is inhibited by the nevirapine (NEV) treatment, capsid and integrase complexes could be observed inside the nucleus (Figure 2). Thus, multiple CA proteins are the shuttle for the viral RNA genome (NEV) or partial/complete retrotranscribed DNA in the host nucleus (absence of NEV). However, these complexes appear to have a different gold distribution from that usually observed in the nucleus of untreated cells [98], which is probably because viruses unable to perform RT remain accumulated in particular nuclear niches [138]. Importantly, these viral complexes are functional because their RT can be restored after drug release and continue viral infection [138]. In particular, we observed that the CA gold labeling distribution changes between the cytoplasm and the nucleus, suggesting a remodeling of the viral core to translocate through the NPC [106]. On the other hand, Zila et al. reported that intact cores can penetrate the NPC and disassemble in the nucleus [78]. As we have discussed, the HIV capsid is an extremely important component of the virus that participates in several steps of the viral life cycle, similarly to the other viruses that we discussed above. In this review, we have also shed light on the importance of the remodeling and structure of viral capsids for the viral infection. It is possible that the HIV capsid is the key component that allows the virus to proceed with an acute or latent infection. HIV-1, likely through the capsid, offers one of the best examples of evolutionary host adaptation that opens a window for better treatments. In particular, it has been observed that mutations in the viral CA do not confer advantages for the viral fitness [139]. This is not good news for the virus, and this viral weakness selects the viral capsid as an exceptional target for a new class of drugs. A recent compound, called Lenacapavir, interferes with the binding of Nup153 and CPSF6 to in vitro assembled cores, stabilizing them [140]. Thus, Lenacapavir is a first-in-class HIV-1 capsid inhibitor with potent antiviral activity against both wild-type virus and resistant mutants to current antiretroviral agents. Lenacapavir is a compound produced by Gilead, and it is already under clinical trial evaluation [141]. However, the race to end the global HIV epidemic is still ongoing. The intriguing discovery of the HIV-induced phase separation involving viral and host components, giving rise to so-called HIV-1 MLOs, could represent a new target step for a new class of drugs to prevent the spread of the virus but also to annihilate the sites where the virus hides.

## 6. Conclusions and Perspectives

Viruses evolved multiple mechanisms to overcome the physical barriers that cell compartments impose for its own protection. This is especially critical for viruses that require access to the nucleus to complete their life cycle. In non-dividing eukaryotic cells, the only door to this compartment is the NPC, which is hijacked by various viral strategies in order to gain access to the nucleus. Some of them such as HBV rely on the manipulation of importins and some nucleoporins such as Nup153 to reach the nuclear compartment. Others viruses such as HSV employ the interaction with nucleoporins of the cytoplasmic side of the NPC, such as Nup214 and Nup358. This interaction helps HSV to inject its genome in the nuclear compartment. On the other hand, the adenoviruses use a combination of binding to the cytoplasmic Nups and to the components of the microtubules to disrupt the capsid and release the genome in the nucleus. Of particular interest is the case of HIV-1 due to the crucial importance of its capsid for nuclear import and the subsequent nuclear steps required for its integration. The HIV-1 virus also hijacks the NPC components located on both sides of the NE (notably Nup358 and Nup153) to efficiently enter the host nuclear compartment. The importance of the capsid in the nuclear import is further highlighted by the observation that even when the retrotranscription is inhibited by nevirapine, the nuclear import still takes place (Figure 2). Once inside the nucleus, the HIV-1 capsid possibly intervenes in the maturation of pre-integration complexes in MLOs (Figure 3).

The nuclear import of viruses is still an unexplored universe with few exceptions that account for a great amount of information such as HIV-1. In many other cases such as adenoviruses, herpesvirus, or HBV, some pioneer studies illustrate novel and interesting aspects of this processes. However, there is much more to discover to generate the complete picture of how different viruses achieve the nuclear access. In the last few years, the continuous optimization of electron microscopy techniques have provided a wealth of information about previously unknown structures and mechanisms employed by viruses. It is likely that in the future, more researchers would have access to these new technologies that combined with more traditional techniques are essential tools to reach new goals in virology. The nuclear import step is essential for establishing a productive infection and viral persistence, so new knowledge on these fundamental aspects of virology will warrant new advances in the field.

## Figures and Tables

**Figure 1 viruses-13-01178-f001:**
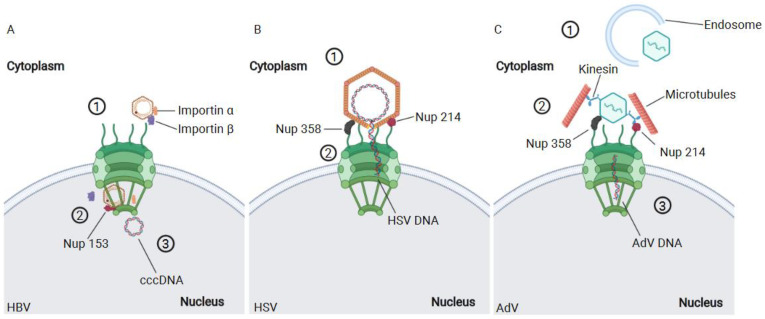
Different viruses employ various strategies in order to access to the nucleus. (**A**) The HBV travels to the vicinity of the NPC protected by its capsid. Thus, it employs importin α and β to access the nucleus (1). The small size of HBV capsid permits the import of this structure complete to the nucleus. The uncoating of HBV capsid takes place assisted by Nup 153 (2) releasing the genome in the nucleus of the cell (3). (**B**) The HSV capsid is too large to be imported complete to the nucleus. Therefore, the HSV binds to the cytoplasmic filaments of NPC via Nup358 and Nup214 (1). Once the capsid docks at the NPC, the viral DNA is injected to the nucleus (2). (**C**) The adenovirus uncoating begins during the release of the viral capsid from the endosomes (1) and is completed at the NPC (2), where there is a complex formed by the capsid, Nup 358, Nup214, microtubules, and kinesin. The mechanical stress caused by the binding to microtubules and to the NPC contributes to completing the tear down of adenovirus capsids. Eventually, the viral DNA is released in the nucleus (3). Created with BioRender.com, accessed on 7 June 2021.

**Figure 2 viruses-13-01178-f002:**
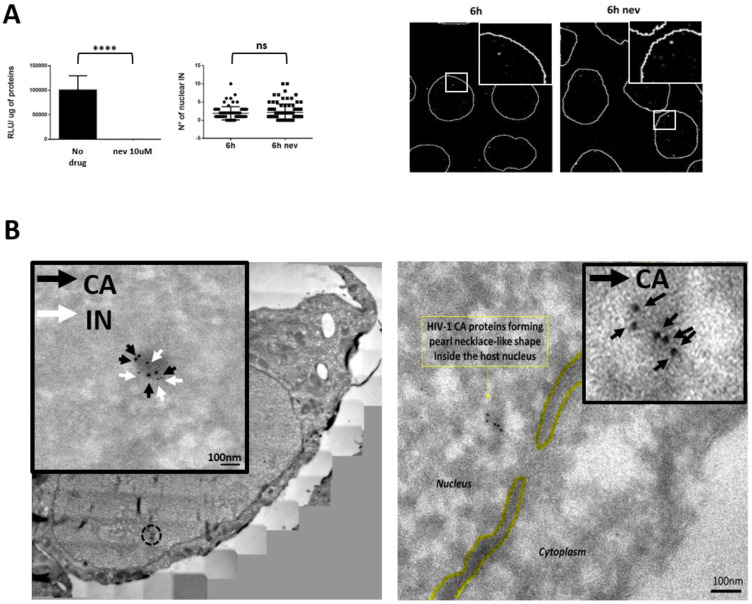
The inhibition of HIV-1 retrotranscription by nevirapine does not block nuclear import of HIV. (**A**) The use of nevirapine (10 µM) efficiently inhibits the reverse transcription and therefore the replication of HIV-1. This blockage does not impede the import of HIV-1 integrase to the nucleus at 6 h post-infection, as the fluorescence microscopy image shows (antibody against the IN-HA tagged has been employed with a secondary antibody conjugated with Alexa 647). (**B**) Transmission electron microscopy (TEM) coupled to immuno-gold labeling (antibodies anti CA and anti IN-HA, respectively recognized by secondary antibodies conjugated with gold particles of 10 nm and 5 nm) reveals the presence of HIV-1 integrase and capsid complexes in the nucleus of infected cells in the presence or in absence of nevirapine. Black arrows represent capsid, whereas integrase is pointed out by the white arrows. *p*-values with <0.0001 are represented by **** than Images adapted from https://www.pasteur.fr/fr/journal-recherche/actualites/technologie-hiv-1-anchor-devoile-vih-1-cellules-vivantes-lors-etape-entree-nucleaire, accessed on 2 June 2021and results corroborating the Blanco-Rodriguez et al. study [106].

**Figure 3 viruses-13-01178-f003:**
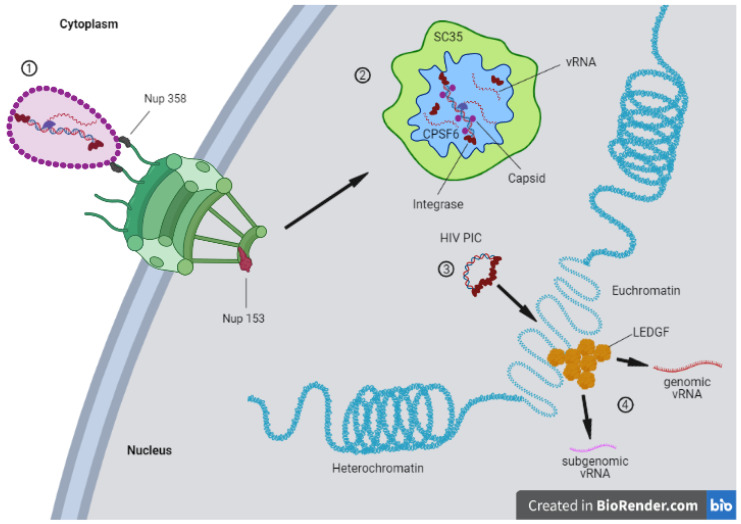
The HIV-1 core carrying the viral genome docks at the NPC followed by nuclear translocation (1). The nuclear RT takes place in HIV-1 MLOs formed by nuclear speckles factors and CPSF6 (2). These are maturation sites of the pre-integration complex (PIC), which once formed is released from them to join euchromatin regions containing LEDGF clusters (3). These are proviral sites where the viral genome is transcribed in genomic and subgenomic viral RNAs (4). Created with BioRender.com, accessed on 7 June 2021.

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
