# Peer review of "The Viral Capsid: A Master Key to Access the Host Nucleus"

_viruses, 2021, doi:10.3390/v13061178_

Round 1

Reviewer 1 Report

In this review manuscript, Blanco-Rodriguez et al. aim at reviewing the recent progresses on the viral-capsid-manipulation of the nuclear import machinery. They chose to discuss how HBV, HSV, Adenovirus and HIV remodel their capsids in order to deliver viral genome into the nucleus. Here are some major issues about this review manuscript:

  • In the section of “Nuclear transport machinery”, it says: “This central region of the NPC 65 contains ~ ten nucleoporins with several phenyl-alanine-glycine that create an unstruc-66 tured mesh in the NPC [25]. FG-nucleoporins from different Nups undergo liquid-liquid 67 phase separation and form liquid droplets that mimic permeability barrier properties of 68 NPCs [26].”. The authors failed to provide a complete and accurate picture regarding the nucleocytoplasmic models and the NPC’s selectivity barrier. There are several different models regarding the configurations of FG-Nup-formed barrier, including hydrogel, polymer brush, ROD, “forest”…Also, there are many experimental and simulation data published to distinguish these models in the past 10-15 years. However, these critical information and progresses are completely missed in this manuscript.
  • It seems that the authors are more familiar with the structural studies by using electron microscopy (EM) and have cited many EM-based publications, however, the functional studies of interactions between viruses and host cells are also very essential for readers to know and should be included in this review manuscript if a complete picture is desired.
  • There are several serious issues about the figures: a) there is no title for figure 1; b) The quality of some figures is too poor to recognize details; c) there is no information if these authors could legally reproduce some figures previously published in other journals.
  • There is no discussion and perspective. This is not acceptable for a review article.

Author Response

We would like to thank the reviewer for his/her suggestions to improve the quality of the document. His/her considerations have been taken into account

  • As suggested new information about the most relevant FG repeats models and supporting experiments have been now included. We expect to offer a more comprehensive view of the selectivity of the NPC with these additions.
  • We have included information about cell/host interaction experiments. We believe our review can now provide an overview of the main findings in viral nuclear import not only obtained from EM studies. Of note, we included a large number of papers and works that contributed to provide a clear view of the biology of viral nuclear import. As suggested by the reviewer.
  • We have corrected the issues that the reviewer suggested. The figure 1 now has a title, the quality of the figures has been improved. Finally, we would like to point out that the figures included in this review are all original and haven’t been published before but supports and describes previous publications.
  • We have included a new section, conclusion and perspectives. We thank the reviewer for this suggestion to increase the quality of the review.

Reviewer 2 Report

The authors review our current knowledge of HBV, HSV, Adenovirus and HIV capsid entry into the nucleus to establish infection.

For the most part the authors do a good job of describing the nuclear pore, shuttling of the viral capsid to the pore and entry into the nuclear space. The images or the figure legends for Figure 2 and Figure 3 are mixed up. The authors should include the Nups associated with HIV capsid entry into the NPC in the cartoon of HIV infections-Figure 2 in text but image Figure 3. 

The authors need to have the manuscript reviewed for English grammar and word usage as this takes away from the quality of the work. 

Author Response

We thank the reviewer for his/her constructive comments. We have updated the review according to his comments.

In the new version we added:

-The correct figure legends

-In figure 3 cartoon now include Nup153 and Nup358 as suggested

-We reviewed the text for English grammar and word usage.